# Measurement report: On the difference of aerosol hygroscopicity between high and low RH conditions in the North China Plain

Jingnan Shi[1,2], Juan Hong[1,2*], Nan Ma[1,2*], Qingwei Luo[1,2], Yao He[1,2], Hanbing Xu[3], Haobo Tan[4,5], Qiaoqiao Wang[1,2], Jiangchuan Tao[1,2], Yaqing Zhou[1,2], Shuang Han[1,2], Long Peng[1,2], Linhong Xie[1,2], Guangsheng Zhou[9], Wanyun Xu[9], Yele Sun[6,7,8], Yafang Cheng[10], Hang Su[10]

[1] Institute for Environmental and Climate Research, Jinan University, Guangzhou, Guangdong 511443, China

[2] Guangdong-Hongkong-Macau Joint Laboratory of Collaborative Innovation for Environmental Quality, Guangzhou, China

[3] Experimental Teaching Center, Sun Yat-Sen University, Guangzhou 510275, China

[4] Institute of Tropical and Marine Meteorology/Guangdong Provincial Key Laboratory of Regional Numerical Weather Prediction, CMA, Guangzhou 510640, China

[5] Foshan Meteorological Service of Guangdong, Foshan 528010, China

[6] State Key Laboratory of Atmospheric Boundary Layer Physics and Atmospheric Chemistry, Institute of Atmospheric Physics, Chinese Academy of Sciences, Beijing 100029, China

[7] College of Earth and Planetary Sciences, University of Chinese Academy of Sciences, Beijing 100049, China

[8] Center for Excellence in Regional Atmospheric Environment, Institute of Urban Environment, Chinese Academy of Sciences, Xiamen 361021, China

[9] Hebei Gucheng, Agrometeorology, National Observation and Research Station, Chinese Academy of Meteorological Sciences, Beijing, 100081, China

[10] Multiphase Chemistry Department, Max Planck Institute for Chemistry, Mainz 55128, Germany

*Correspondence to*: Juan Hong (juanhong0108@jnu.edu.cn) and Nan Ma (nan.ma@jnu.edu.cn)

**Abstract.** Atmospheric processes, including both primary emissions and secondary formation, may exert complex effects on aerosol hygroscopicity, which is of significant importance in understanding and quantifying the effect of aerosols on climate and human health. In order to explore the influence of local emissions and secondary formation processes on aerosol hygroscopicity, we investigated the hygroscopic properties of submicron aerosol particles at a rural site in the North China Plain (NCP) in winter 2018. This was conducted by simultaneous measurements of aerosol hygroscopicity and chemical composition, using a self-assembled hygroscopic tandem differential mobility analyzer (HTDMA) and a capture-vaporizer time-of-flight aerosol chemical speciation monitor (CV-ToF-ACSM). The hygroscopicity results showed that the particles during the entire campaign were

mainly externally mixed, with a more hygroscopic (MH) mode and a less hygroscopic (LH) mode. The mean hygroscopicity parameter values ($\kappa_{mean}$) derived from hygroscopicity measurements for particles at 60, 100, 150, and 200 nm were 0.16, 0.18, 0.16, and 0.15, respectively. During this study, we classified two distinct episodes with different RH/$T$ conditions, indicative of different primary emissions and secondary formation processes. It was observed that aerosols at all measured sizes were more hygroscopic under the high RH (HRH) episode than those under the low RH (LRH) episode. During the LRH, $\kappa$ decreased with increasing particle size, which may be explained by the enhanced domestic heating at low temperature, causing large emissions of non- or less-hygroscopic primary aerosols. This is particularly obvious for 200 nm particles, with a dominant number fraction (> 50 %) of LH mode particles. Using O : C-dependent hygroscopic parameters of secondary organic compounds ($\kappa_{SOA}$), closure analysis between the HTDMA measured $\kappa$ and the ACSM derived $\kappa$ was carried out. The results showed that $\kappa_{SOA}$ under the LRH episode was less sensitive to the changes in organic oxidation level, while $\kappa_{SOA}$ under the HRH had a relatively stronger dependency on the organic O : C. This feature suggests that the different sources and aerosol evolution processes, partly resulting from the variation in atmospheric RH/$T$ conditions, may lead to significant changes in aerosol chemical composition, which will further influence their corresponding physical properties.

## 1    Introduction

The hygroscopicity of aerosol, describing its tendency to absorb moisture from the environment, plays an important role in understanding and quantifying its effects on climate and human health (Martin et al., 2004; McFiggans et al., 2006; Swietlicki et al., 2008; Tao et al., 2012). Hygroscopic growth alters the global radiative balance directly by modifying the size distribution of aerosol particles and consequently influencing their light scattering and adsorption coefficients (Cheng et al., 2008; Sloane and Wolff, 1985). Hygroscopicity is also closely tied to the activation of particles to form cloud droplets, thus may influence the lifetime and microproperties of clouds, affecting the regional and global climate indirectly (Gunthe et al., 2009; Rose et al., 2010). Moreover, due to the absorbed water by aerosol particles, the hygroscopicity of particles is strongly associated with heterogeneous and multiphase chemistry, regulating the abundance of different species in the gas and particle phases (Hennigan et al., 2009; Ye et al., 2011). Furthermore, aerosol hygroscopicity can also influence the

human health by changing the deposition of aerosol particles in the respiratory system (Kreyling et al., 2006).

According to the Köhler theory, the propensity of aerosols to uptake water at a certain sub/supersaturation is mainly determined by the particle size and various properties of the species within the particles, such as solubility, molecular weight, density, that is to say, their chemical composition (Petters and Kreidenweis, 2008, 2007). Thus, ambient aerosols owing to their different

sources and atmospheric processes, may vary greatly in their chemical compositions and thus show significant difference in their hygroscopicity (Asmi et al., 2010; Cai et al., 2017; Ehn et al., 2007; Fan et al., 2020; Hong et al., 2015). Previous studies have shown that primary emitted aerosol, which normally contains substantial insoluble components, tends to have a lower hygroscopicity. Cai et al. (2017) measured the hygroscopicity at a suburban site in the Pearl River Delta (PRD) in China and a

marine site in Okinawa, Japan. They found that the Okinawa aerosols were much more hygroscopic than that of PRD aerosols, which had a stronger influence from the traffic-related exhaust and other directly emitted pollutants. Fan et al. (2020) showed that the hygroscopicity of aerosol in Beijing was lower in winter than that in summer, owing to the larger contribution of less hygroscopic material from coal-burning and other primary emissions in the winter-time of Beijing. There are also some other

studies, which focused on the secondary formed or aged aerosols, being typically characterized as more hygroscopic. Asmi et al. (2010) measured the hygroscopicity of aerosols in the Antarctica region and concluded that the Antarctica aerosols with such a strong hygroscopicity was associated with the aging of secondary inorganic aerosols. Hong et al. (2015) reported that the aerosols at a boreal forest in Southern Finland, representing typically the secondary biogenic sources, were quite hygroscopic,

especially during the daytime when the photochemistry was stronger. All the above findings imply that due to the different types of primary sources and complexity of secondary formation processes in different regions, the understanding of the connections between aerosol sources as well as its evolution processes and its hygroscopicity is still limited.

The North China Plain (NCP), one of the most populated and developed regions in China (Yang et al., 2018), has suffered from severe air pollution over the past decade (Yue et al., 2011; Zhang et al., 2016).

Due to the intensive human activities, large amounts of primary air pollutants were emitted to the environment in this region, especially during winter time (Du et al., 2018; Fan et al., 2020). Coupled with the unfavorable meteorological conditions in the winter-time of NCP, this further led to a fast production of secondary aerosols (Sun et al., 2013). Consequently, particle phase chemical composition of this region showed distinct feature owing to the complex contribution from both primary emissions and secondary formation, which might result in obvious differences in aerosol hygroscopicity. Previous studies (Massling et al., 2009; Liu et al., 2014; Wu et al., 2016; Qi et al., 2018; Wang et al., 2018d; Shen et al., 2021) have devoted extensive efforts to investigate the hygroscopicity of aerosols in the NCP. However, these studies mainly focused on the statistical analysis of the aerosol hygroscopicity in this region and tried to investigate its characteristics from the point of view of particle phase chemical composition, but seldom link to their sources or formation processes. Thus, the representative feature that how different sources, including both the primary emissions and secondary processes, impact the hygroscopicity of NCP aerosols still remains unclear.

In this study, we aim to explore the effects of different atmospheric processes on the aerosol hygroscopicity in the NCP. For this purpose, we measured the hygroscopicity of atmospheric aerosols at a rural measurement station in the NCP with a self-assembled HTDMA system. We compared the characteristic hygroscopicity of ambient aerosols during different episodes in which aerosols were dominated by either primary sources or secondary formation, in order to reveal the impact of different atmospheric processes/sources on aerosol hygroscopicity.

## 2    Experiment

### 2.1 Measurement site

Hygroscopicity measurements and chemical characterization of atmospheric aerosols were conducted at a rural measurement station (the Ecological and Environmental Monitoring Station of the Chinese Academy of Meteorological Sciences) in Gucheng, Dingxing County, Hebei Province (39.15° N, 115.74° E, 22 m a.s.l.) from November 11th to December 14th in 2018. The observational site is surrounded by farmland and small residential towns with a national highway, about one kilometer away to the west. The nearest large cities are Tianjin (110 km) and Beijing (100 km), two megacities in the

NCP. Measurements at this site could well represent the regional conditions of the NCP. Detailed descriptions of the site can be found in Li et al. (2021). During the campaign, ambient meteorological conditions including air pressure, temperature, RH, wind speed, wind direction and precipitation were measured continuously at the station. All the instruments for aerosol measurements were operated in a container, where temperature was maintained at 24 °C, and connected to a sampling inlet system using an isokinetic flow splitter. The sampling air was dried to RH below 20 % using a Nafion diffusion dryer.

## 2.2 Hygroscopicity measurements

In this study, the hygroscopic growth of atmospheric particles was measured with a self-assembled HTDMA system. The detailed description of the HTDMA system could be found in previous studies (Hong et al., 2018; Tan et al., 2013). Briefly, ambient particles were dried to RH lower than 20 % by passing through a Nafion dryer (Model PD-70T- 24 ss, Perma Pure Inc.). After charging to a known charge distribution through a $Kr^{85}$ neutralizer (TSI, 3077), these particles were introduced into a differential mobility analyzer (DMA1, model 3081; TSI, Inc), where particles at a certain size were selected. Then these quasi-monodisperse particles were humidified to a specific RH (RH = 90 % in this study) in a Nafion humidifier (Perma Pure, PD-100T-24MSS). The residence time of the humidification unit is around 3 seconds. With a second DMA (DMA2, Model 3081L, TSI Inc.) and a condensation particle counter (CPC, model 3772; TSI, Inc), operated at a flow rate of 1 L min$^{-1}$, the number size distribution of the humidified aerosols was measured. The hygroscopic growth factor (HGF) was then obtained, which is defined as (Eq. 1) the ratio of the particle diameter at a given RH ($D$ (RH), RH = 90 %) to the dry diameter ($D_0$):

$$HGF = \frac{D(RH)}{D_0} ,\qquad\qquad(1)$$

In this study, particles of four different mobility diameters ($D_0$ = 60, 100, 150, and 200 nm) were measured with a time resolution of 20 minutes. Raw data were inverted by the TDMA$_{inv}$ algorithm designed by Gysel. (2009). Polystyrene latex spheres (PSL, Thermo Scientific, Duke Standards) were used to verify the sizing accuracy of both DMAs in the system. Calibration measurements in terms of the accuracy and stability of RH inside the HTDMA were performed using ammonium sulfate particles, of which the deliquescence RH and hygroscopic growth factor at different RHs are well known.

Calibration measurements using ammonium sulfate were conducted once a week.

## 2.3 Chemical composition measurements

An Aerodyne capture-vaporizer time-of-flight aerosol chemical composition monitor (CV-ToF-ACSM, ACSM hereafter) was used to measure the non-refractory $PM_1$ chemical composition, including $NH_4^+$, $SO_4^{2-}$, $NO_3^-$, $Cl^-$, and organics at a time resolution of 2 min (Zheng et al., 2020). The instrument was operated at a flow rate of 0.1 L min$^{-1}$ with a 100 % collection efficiency during the sampling period. The oxidation level (O : C ratio) of the organic fractions were estimated by the mass fractions of $m/z$ 44

($f44$) in total organics according to Canagaratna et al. (2015). In order to further characterize the origin and evolution of the organic fractions in aerosols (Ng et al., 2010), subsequent analysis of OA mass spectra obtained from ACSM measurements were deployed using positive matrix factor (PMF) (Paatero and Tapper, 1994; Ulbrich et al., 2009). Several OA factors corresponding to different sources and processes, including hydrocarbon-like OA (HOA), cooking OA (COA), coal combustion OA

(CCOA), and biomass burning OA (BBOA), as well as oxygenated OA (OOA) were identified. In particular, HOA, COA, CCOA and BBOA were typically considered as primary factors (POA) and OOA as secondary factor (SOA). Detailed descriptions of the measurements and PMF analysis can be found in Sun et al. (2018).

The mass concentration of elemental carbon (EC) and organic carbon (OC) were analyzed using an OC/EC aerosol analyzer (Sunset Laboratory, Forest Grove, OR) (Bae et al., 2004) through a $PM_{10}$ inlet. A detailed description of the instrument can be found in Jeong et al. (2004). The EC (thereafter BC in our study) mass concentration in $PM_1$ was determined from the measured EC in $PM_{10}$ using a correction factor according to Chen et al. (2019).

## 3   Methedology for data analysis

### 3.1 Hygroscopicity parameter

According to the $\kappa$-Köhler theory (Petters and Kreidenweis, 2007), the hygroscopic parameter ($\kappa$) describes the water uptake ability of particles, which can be calculated at any given RHs or supersaturation. From the HTDMA measurements, $\kappa$ can be derived using the following equations:

$$\kappa_{HTDMA} = (HGF^3 - 1)\left(\frac{\exp\left(\frac{A}{D_0 HGF}\right)}{RH} - 1\right), \tag{2}$$

$$A = \frac{4\sigma_{s/a}M_w}{RT\rho_w}, \tag{3}$$

where HGF is the hygroscopic growth factor measured by our HTDMA at a given RH (e.g., 90 % in our study), $D_0$ is the size of particles selected by the first DMA, $\sigma_{s/a}$ is the droplet surface tension, which was assumed to be the surface tension of pure water ($\sigma_{s/a} = 0.0728N\ m^{-2}$), $Mw$ is the molecular weight of water, $R$ is the universal gas constant, $T$ is the absolute temperature, and $\rho_w$ is the density of water. According to the derived $\kappa$ distribution, we defined two hygroscopic groups for the ambient particles measured in our study: $\kappa < 0.1$ as particles of less hygroscopic (LH) mode and $\kappa > 0.1$ as more hygroscopic (MH) particles (Liu et al., 2011).

### 3.2 Hygroscopicity- chemical composition closure

For ambient mixed particles, $\kappa$ can also be predicted by the Zdanovskii Stokes Robinson (ZSR) mixing rule assuming volume additivity. Specifically, $\kappa$ of a mixed particle ($\kappa_{mix}$) (Eq. 4) can be calculated by using the hygroscopicity parameter of the components in the mixture and their respective volume fractions (Petters and Kreidenweis, 2007),

$$\kappa_{mixed} = \sum_i \kappa_i \times \varepsilon_i, \tag{4}$$

where $\kappa_i$ is the hygroscopicity parameter of a component $i$, and $\varepsilon_i$ is the volume fraction of the component $i$ in the mixed particles which can be derived using particle chemical composition data measured by ACSM and OC/EC. The same approach to calculate $\varepsilon_i$ based on particle chemical composition was described in previous studies with more details (Hong et al., 2014, 2018). The volume fractions of six neutral components, including $(NH_4)_2SO_4$, $NH_4HSO_4$, $NH_4NO_3$, $H_2SO_4$, organics and BC were ultimately determined and their corresponding $\kappa$ measured by previous works (Cross et al., 2007; Gysel et al., 2007; Wu et al., 2016) were summarized in Table 1. Note that the hygroscopicity parameter for organic species ($\kappa_{org}$) was not well characterized in contrast to inorganic salts. Alternatively, $\kappa_{org}$ was typically assumed to be constant ranging from 0.1 to 0.4 or varied as a function of their oxidation level when predicting $\kappa_{mix}$ in previous works (Asmi et al., 2010; Chang et al., 2010; Gunthe et al., 2009). In our case, a distinct approach to approximate the hygroscopicity of organics ($\kappa_{org}$), particularly for that of SOA ($\kappa_{SOA}$) was introduced, based on the aforementioned material and

method. Detailed description on the approximation of $\kappa_{org}$ will be given in Sect. 4.4, and therefore $\kappa_{org}$ in Table 1 was not specified.

## 4    Results and discussion

### 4.1  Overview of the measurement

Figure 1 displays the time series of meteorological conditions (e.g., the average wind speed, wind direction, temperature, and relative humidity), mass concentration of $PM_{2.5}$ and BC, the averaged hygroscopicity parameter ($\kappa$) and $\kappa$ probability density function ($\kappa$-PDF) for particles at dry sizes of 60, 100, 150 and 200 nm and the mass fractions of chemical components in $PM_1$ measured by ACSM, respectively. In general, the wind speed was typically quite low during the nighttime compared to that during daytime, leading to a limited dilution of the air pollutants in the rural atmosphere. Thus, a similar diurnal pattern was also observed for BC (ranging from 0 to 30 $\mu g \ m^{-3}$) and $PM_{2.5}$ (ranging from 1.72 to 245.14 $\mu g \ m^{-3}$) mass concentration. The ambient temperature ($T$) increased during the daytime with a relative low RH and dropped dramatically during the nighttime, leading to a significant enhancement of ambient RH. Clearly, we observed two different episodes with distinguished difference in ambient RH and $T$, during which the contribution of primary aerosols and secondary aerosol formation processes might vary significantly. Thus, two distinct episodes (e.g., "high RH (HRH) episode" on 19–27 November and "low RH (LRH) episode" on 7–14 December) were characterized. To be specific, the average RH during the HRH episode was 71 % ± 22 %, with an average temperature of 3 °C, while during the LRH episode the average RH was 43 % ± 17 %, with an average temperature of -6 °C. Separate analysis of the hygroscopicity and chemical composition of aerosols for these two episodes will be further discussed in Sect.4.2-4.4.

The averaged hygroscopicity parameter $\kappa$ for particles at each size (black lines in Fig. 1d-g) shows a clear temporal variation, which is similar to the mass fraction of chemical composition in $PM_1$. Two distinct modes with $\kappa < 0.1$ as less hygroscopic (LH) mode and $\kappa > 0.1$ as more hygroscopic (MH) mode were mostly observed for all sized particles from the $\kappa$-PDF, indicating that the particles were mainly externally mixed during our measurements. This was further confirmed by the clear bimodal distribution from the averaged $\kappa$-PDF in Fig. 2. Both the averaged and the time series of $\kappa$-PDF shows

a strong size dependency. For instance, the number fraction of 60 nm particles was dominated by the MH mode during most of time. However, with increasing particle size, the LH mode became more prominent. This is particularly obvious for 200 nm particles with a clear dominance of LH particles, especially during the LRH episode.

During our study, the ensemble mean hygroscopicity parameter ($\kappa_{mean}$) were 0.16, 0.18, 0.16, and 0.15 for 60, 100, 150, and 200 nm particles, respectively, as shown in Fig. 3. The results in our study show that the aerosols at current station have the lowest hygroscopicity compared with aerosols in other cities or regions in China. This is more likely due to the largest contribution of organics relative to inorganic species in $PM_1$ at our observational site. Slightly higher hygroscopicity was observed for the aerosols in other regions of the NCP, for instance, the urban aerosols in Beijing (Wang et al., 2018a), suggesting similar primary emissions or secondary aerosol formation processes in the NCP. Moreover, a clear increase trend in $\kappa_{mean}$ with increasing particle size was observed in most of previous studies in China (Jiang et al., 2016; Wu et al., 2013; Ye et al., 2011; Yuan et al., 2020). However, on the contrary, our particles at larger sizes were less hygroscopic than smaller ones. This phenomenon was also observed for the aerosols in the wintertime of Beijing in 2005 and 2015 (Massling et al., 2009; Wang et al., 2018a). Detailed explanation with respect to the lower hygroscopicity at larger size will be given in the following section.

**4.2 Different aerosol hygroscopicity under the HRH and LRH episodes**

In order to explore the effects of different atmospheric emissions/processes on the aerosol hygroscopicity in the NCP, we analyzed the hygroscopicity and mixing state of particles at different sizes during the two distinguished episodes, respectively. Figure 4 shows the size-dependent number fractions of LH mode ($NF_{LH}$) and MH mode ($NF_{MH}$) particles under these two distinct episodes. It can be seen that under the HRH episode, $NF_{LH}$ increased slightly with increasing particle size (e.g., from 20 % for 60 nm particles to 30 % for 200 nm particles), with MH mode still being the dominant. Under the LRH episode, however, a substantial increase in $NF_{LH}$ with particle size (e.g., from 28 % to 53 %) was observed with an obvious decrease in $NF_{MH}$ in large particles accordingly. As a result, the average $\kappa$ values for each size, indicated in Fig. 4b, were generally lower under the LRH episode than those

under the HRH episode. Moreover, under the LRH episode, an obvious lower $\kappa$ was observed for larger particles (e.g., 200 nm particles) compared to that of smaller ones, which is probably due to the higher

content of LH mode particles in aerosols (see Fig. 4d).

The clear variability in aerosol hygroscopicity between these two episodes is not surprising that at our observational site organics were the dominant species in $PM_1$ under the LRH episode, while inorganic species dominated the $PM_1$ under the HRH, as shown in Fig. 5. It has to be noted that the

submicrometer mass concentration illustrated in Fig. 5 represented the $PM_1$ bulk chemical composition, which was normally dominated by particles at the size near the mode size in the mass size distribution. Therefore, we further analyzed our particle number size distribution (7 nm – 9701 nm) data and found that the mode size of the mass size distribution of aerosols during our experimental campaign was around 390 nm, assuming a constant particle density of 1600 $k$g m$^{-3}$. Based on this result, we

considered that the bulk chemical composition measured by our ACSM could nearly reflect the ones for particles near the mode size, for instance, 200 – 300 nm.

Back to the discussion on the particle chemical composition, the different content of organic and inorganic species in particles reveals distinct sources or formation mechanisms of these components

under different episodes. This in turn influences the aerosol hygroscopicity, as aerosols with more organics that typically have a lower $\kappa$ than inorganic species will be less hygroscopic than those having more inorganics. Inorganic species, i.e., sulfate, nitrate, and ammonium are usually considered to be mainly from secondary formations (Sun et al., 2015). In our study, inorganic species contributed 44 % to the $PM_1$ mass in the HRH episode, which were higher than those under the LRH episode. This was

especially obvious for nitrate, which accounted for 19 % of mass in $PM_1$ under the HRH but only 9 % under the LRH. Previous studies (Li et al., 2018; Liu et al., 2020; Wang et al., 2018) have found that ambient conditions with substantial nitrate in ambient aerosols were typically associated with elevated ambient RH conditions, which would lead to a higher aerosol liquid water content. This would in turn promote the reactive uptake of precursors and thermodynamic equilibrium of ammonium nitrate and

thus result in an increased particulate nitrate formation (Tong et al., 2020; Wang et al., 2020). These findings may be responsible for the significant increase in nitrate content in aerosols during the HRH

episode in our study. As nitrate is quite hygroscopic, this feature may be one of the most important reasons for the obvious difference in aerosol hygroscopicity between these two episodes, at least for 200 nm particles.


Organic fraction, which is another major component in aerosols, also varies significantly in hygroscopicity due to the exist of numerous and highly diverse organic compounds. Based on this, we grouped them into several individual factors according to their potentially similar sources under these two different episodes, as illustrated in the bottom panel of Fig. 5. We can see that the mass fraction of

POA in the total organics during the HRH episode was 23 % lower than that during the LRH episode. In particular, CCOA, which is the main component of POA, accounted only for 6 % of the total organics under the HRH episode, but as much as 29 % under the LRH. Considering the distinct content of total organics between the two episodes, the difference in the mass fraction of CCOA in $PM_1$ would be even more significant, being 2.8 % (46 %×6 %) for the HRH and 17% (60 %×29 %) for the LRH.

The high level of CCOA in aerosols could be explained that during the LRH episode, the wintertime residential heating was initiated, during which a significant amount of its pollutants, for instance, CCOA, was emitted to the atmosphere (Hua et al., 2018). Moreover, we observed that the mass fraction of OOA in $PM_1$ during the HRH episode was 19% (46 %×41 %), being different from that (11%, e.g., 60 %×18 %) during the LRH, though only slightly higher. Thereby, the varied level of different

organics in $PM_1$ could be one of the plausible reasons for the different $\kappa$ in 200 nm between these two episodes.

Moreover, we observed that both the mass fraction and the absolute mass concentration of OOA during the HRH episode was higher than that during the LRH. Kuang et al. (2020) specifically studied the

formation mechanism of secondary organic aerosols for current campaign. They found that the daytime OOA formation rates correlated quite well with nitrate formation rates and thus suggested that they probably shared similar formation pathways. They further investigated the formation pathways of nitrate and concluded that during the low RH conditions, nitrate was mainly formed by the gas-phase oxidation of $NO_2$, while at the high RH conditions, both the aqueous-phase processes and gas-phase

oxidation dominated its formation. Furthermore, they considered that gas-phase formation of OOA

would mainly add mass to the condensation mode of aerosol size distribution, while aqueous-phase formation of OOA may elevate the mass in the droplet mode. After studying the diurnal evolution of aerosol mass distribution for high RH and low RH conditions, together with the indirect evidence from nitrate formation, they concluded that gas-phase formation contributed dominantly to the OOA under the low RH conditions, while at high RH conditions, when aerosol water content was high, aqueous-phase photo-oxidation were mainly responsible for the rapid OOA formation, beside the gas-phase formation. According to their further comparison with laboratory experiments (Sun et al., 2010; Yu et al., 2014), they suggested that the OOA formed through aqueous-phase reactions were normally more-oxidized compared to that formed by gaseous processes. It needs to specify that the average RH values for the HRH and LRH episode during our campaign were 71 % and 43 %, respectively, being quite close to the ones (53 % & 38 %) for the two defined RH conditions in Kuang et al. (2020). Taking all together, this could reasonably to presume that the observed lower $\kappa$ for 200 nm particles under the LHR episode, not only due to the elevated POA fraction in aerosols, but also a different hygroscopic nature of OOA, compared to that under HRH. All these above features with respect to both inorganic and organic fractions demonstrate that the complexity in primary and secondary aerosol formation processes under these two distinct episodes led to a particular variability in aerosol chemical composition, which further resulted in different characteristics in aerosol hygroscopicity.

### 4.3 Diurnal variation

To better understand the influence of human activities and secondary formation on the aerosol hygroscopicity of current study on a daily scale, we compared the diurnal variation of the number fractions and individual $\kappa$ of LH and MH mode particles under these two episodes, as shown in Fig. 6. Under the HRH episode, particles at all four sizes (60 nm, 100nm, 150 nm, 200 nm) were dominated by the MH mode, which has been already discussed in Sect. 4.2. The general pattern for the number fraction of LH mode particles under the HRH episode did not vary significantly among particles of different sizes, with a sharp decrease after 07:00 in the morning and a slow increase around noon till 20:00 in the evening. Similar trends were also observed for particles at 100, 150 and 200 nm under the LRH episode except 60 nm particles. Concurrently, the obtained $\kappa$ of LH mode particles at all four sizes,

no matter which episode was considered, increased slightly in the morning, reached the peak around

noon and decreased back until the evening.

The higher hygroscopicity of LH mode particles and lower degree of external mixing during daytime (except for the 60 nm particles under the LRH episode) could be explained by that these LH mode particles started to age in the atmosphere during the daytime either by the photochemistry processes or

multiphase reactions. These aged particles became more hygroscopic and thus the increase in $\kappa$ were observed in the daytime for LH particles. However, part of these aged LH particles become much more hygroscopic, with the obtained $\kappa$ beyond the range for LH mode but reach to that of MH mode particles. Therefore, these particles were classified into MH fractions and thus a lower number fraction of LH particles was observed.


The elevated hygroscopicity during daytime for MH mode particles was also observed, especially apparent during the HRH episode, except for that of 60 nm particles, as illustrated in Fig. 6e. During daytime when solar radiation is stronger, higher concentration of atmospheric oxidant was typically observed (Hong et al., 2015). Thus, the associated photochemistry and other aging processes would be

more favorable, leading to a more oxidized material condensing or partition into the existing aerosols and thus an elevated aerosol hygroscopicity being expected.

Particularly for 60 nm particles, both the number fraction of LH mode particles under the LRH episode and the hygroscopicity of MH mode under the HRH show a different diurnal trend compared to that of

particles at the other three sizes. Higher fractions of their LH mode particles were observed around afternoon, peaking at around 15:00. This indicates a different source for the LH group of Aitken mode particles, for instance, residential heating as well as local traffic emissions. The diurnal cycle of the POA mass fraction in Fig. 7 were consistent with those for the number fractions of LH mode particles at larger sizes. As POA are non- or less-hygroscopic, this further suggests that the likely candidate for

these LH mode particles could be soot or water-insoluble organics, for instance, similar component as POA. On the other hand, the hygroscopicity of MH mode particles decreased in the daytime under the HRH episode, opposite to the other sizes. The reason for this pattern is not quite clear and indicates a

very different atmospheric process of these Aitken mode particles, which should be analyzed further in future work.

The daily variations of all the factors, including the relative number fraction of LH/MH mode particles and their corresponding hygroscopicity, contribute together to the diurnal pattern of the overall ensemble mean $\kappa$ for particles at each size, with generally higher values under the HRH compared to that under the LRH episode (see Fig. 6g-h). This pattern is reasonable as there were more inorganic species as well as OOA in particles under the HRH episode (Fig. 7), their observed mean $\kappa$, at least for

larger particles, was supposed to be larger compared to that under the LRH.

**4.4 Relating the SOA hygroscopicity to its oxidation level for the two episodes**

As introduced in Sect. 3.2, the hygroscopicity of particles can also be derived on the basis of chemical composition measurements according to the ZSR mixing rule. The $\kappa_{\text{org}}$ is the biggest unknown in the composition based derivations and the different choices $\kappa_{\text{org}}$ could lead to large deviations in the

hygroscopicity for the studied mixtures. This is particularly difficult for secondary organics, as POA was normally considered to be non-hygroscopic. However, coupling HTDMA measured $\kappa$ with particle phase chemical composition data, we may be able to approximate the hygroscopicity for organics, especially the secondary organics. Specifically, $\kappa_{\text{SOA}}$ can be calculated by substracting $\kappa$ of inorganic species, POA and BC from HTDMA measured $\kappa$ as (Eq. 5):

$$\kappa_{SOA} = \frac{\kappa_{HTDMA} - \kappa_{Inorg} \times \varepsilon_{Inorg} - \kappa_{BC} \times \varepsilon_{BC} - \kappa_{POA} \times \varepsilon_{POA}}{\varepsilon_{SOA}},$$ (5)

where $\kappa_{Inorg}$ represent $\kappa$ of all the inorganic species, POA and BC measured by ACSM assuming ZSR mixing rule and $\varepsilon_i$ is the volume fraction of different components. As POA and BC is not hygroscopic, $\kappa_{POA}$ and $\kappa_{BC}$ of 0 was assumed. This approach for the approximation of $\kappa_{SOA}$ has also been applied in previous studies (Wu et al., 2016, 2013). As the bulk chemical composition measured by ACSM may

deviate significantly from that of size-resolved ones, we plotted the particle mass distribution of aerosols averaged over the entire campaign, see Fig. S1 in the supplement. From Fig. S1, we found that the mode size of the mass size distribution of aerosols during our experimental campaign was around 390 nm. Thereby, we considered that the bulk chemical composition measured by our ACSM could nearly reflect or at least be close to that of 200 nm particles.


Previous studies suggested that the hygroscopicity of organic components varies linearly with their oxidation level (O : C). In order to investigate the different characteristic of secondary organic aerosols during these two distinct episodes, we separately examined the relationship between the organic hygroscopicity ($\kappa_{SOA}$) and their oxidation level. Figure 8 shows the positive correlations between $\kappa_{SOA}$

and O : C ratios for organics during both episodes but with different O : C dependencies. It is important to note that even though we considered the chemical composition of 200 nm particles was close to that of $PM_1$ measured by ACSM, there still potentially remained deviations between size-resolved chemical composition and bulk ones (Xu et al., 2021). Therefore, current correlations were only used for characteristic comparison of organic hygroscopicity between these two episodes. Parameterization

using these correlations for organic aerosols at current site from a broader perspective, for instance, for climate modelers, is not suggested.

Specifically, under the HRH episode, the O : C ratio of organic components ranged between 0.30 to 1.29 with an average value of 0.76. It was noted that due to the availability of hygroscopicity data, not

all of the measured O : C values were shown and a smaller data coverage was displayed in Fig. 8. The hygroscopicity of secondary organic aerosols during this episode shows a relatively stronger O : C dependency as $\kappa_{SOA} = 0.17 \times$ O : C + 0.04. Considering this relation, the average O : C ratio of this episode corresponds to a mean $\kappa_{SOA}$ of 0.17, which is moderately hygroscopic. This feature is similar to the suburban organic aerosols in Guangzhou measured by Hong et al. (2018). The average RH during

their study in Guangzhou was 72 %, being also quite high and thus suggesting potentially similar formation mechanism for secondary organic aerosols, e.g., aqueous-phase photochemistry as speculated previously.

In contrast, the hygroscopicity of SOA during LRH episode shows a weak O : C dependent as $\kappa_{SOA} =$

$0.09 \times$ O : C + 0.03. Considering that the average O : C ratio was 0.52 over the entire LRH episode, spreading from 0.18 to 1.24, the average $\kappa_{SOA}$ of the LRH organics was approximately 0.08, much lower than that under the HRH. Similarly, Wu et al. (2016), performed in the urban area of the NCP, showed marginal increase in hygroscopicity with the O : C ratio, approximate to the ones for our LRH aerosols. We found that during Wu et al. (2016)' study, ambient RH was 40 % on average. Thereby, we

speculate that the similar characteristic in hygroscopicity of secondary organic aerosols in the NCP of both studies was likely due to the comparable low RH conditions, indicative of similar formation pathways. Moreover, the difference in $\kappa_{SOA}$ under these two episodes is physically reasonable that under the HRH episode, as previously suggested, the rapid OOA formation were mainly produced through aqueous-phase photo-oxidation, which tends to form more-oxidized organics. On the contrary,

relatively less-oxidized organics, which were less hygroscopic, were mostly formed through gas-phase reactions under the LRH episode.

As the ambient RH had a large diurnal variation during our campaign, which implies that low/high RH conditions may also be occur during any individual day of the HRH/LRH episode, we further grouped

the data points of Fig. 8 into two categories according to their absolute RH values for these two episodes, as shown in Fig. 9. The threshold at RH of 60 % was set for these two categories due to the RH intensity spread in Fig. 8. At conditions of RH larger than 60 %, the hygroscopicity of SOA under the HRH and LRH episodes both show a strong O:C-dependency, with the fitting under the LRH being more skewed. At RHs lower than 60%, the relationship between $\kappa_{SOA}$ and the O:C for these two

episodes became even more closer. However, we observed that the absolute value of $\kappa_{SOA}$ still varied between these two episodes, even at similar RH ranges, though their individual behavior towards the variation of O:C was similar. This implies that there still remain differences in these SOAs at different episodes but similar RH conditions, for instance, their chemical composition, indicating that the formation pathways of these SOA or the relevant reaction precursors might still be different under these

two episodes. Thus, separation of these two episodes as previously defined was kept as the main conclusion was not altered and distinct groups with respect to their RH will not be merged further.

**5 Conclusions**

Simultaneous measurements of hygroscopicity and chemical fractions were performed at a rural observational site of the NCP during the winter of 2018. The hygroscopic probability density function

($\kappa$-PDF) distribution indicates that aerosols in this region were usually externally mixed. Particle hygroscopicity decreased with increasing aerosol size, with $\kappa_{mean}$ values of 0.16, 0.18, 0.16, and 0.15 for 60 nm, 100 nm, 150 nm and 200 nm particles, respectively. Different RH/T conditions, indicating

distinctive primary emissions and secondary aerosol formation processes, were distinguished, namely the HRH episodes and the LRH episodes, respectively. The mean $\kappa$ value of particles at each size was generally lower under the LRH episode than that under the HRH episode. During the HRH episode, $NF_{LH}$ increased slightly with increasing particle size, while under the LRH episodes, this trend was opposite with the LH mode particle being the dominant for 200 nm particles. The different contribution of less hygroscopic material from primary emissions and the complexity in the secondary aerosol formation, such as the formation of nitrate and OOA, were the main reasons for the distinct hygroscopicity of aerosols between the two episodes.

The HTDMA measured $\kappa$ were compared against the ones predicted by ACSM to obtain a relationship between the $\kappa_{SOA}$ and their oxidation level (O : C). The results show that $\kappa_{SOA}$ under LRH conditions was less sensitive to the changes in organic oxidation levels, while $\kappa_{SOA}$ under HRH conditions had a stronger O : C-dependency. Based on these findings, we concluded that intricate atmospheric processes/emission sources could exert significant influence on the chemical composition of atmospheric aerosols, leading to synthetic variation of aerosol hygroscopicity. The variability of aerosol hygroscopicity, on the other hand, may in turn benefit our understanding of the formation and evolution of complex atmospheric processes.

**Declaration of interest statement.**

The authors declare that they have no known competing financial interests or personal relationships that could have appeared to influence the work reported in this paper.


**Data availability.**

The details data can be obtained from https://doi.org/10.5281/zenodo.5758744 (Shi, 2021).

**Author contributions.**

JNS analyzed the data, plotted the figures and wrote the original draft. JH planned the study, collected the resources, wrote and finalized the manuscript, NM collected the resources, reviewed and finalized the manuscript. HBX contributed to the instrument maintenance, HBT, QWL, YH, JCT, YQZ, LP, LHX, GSZ, WYX, YLS and SH discussed the results. QQW, YFC and HS contributed to fund acquisition.

**Competing interests.**

Hang Su and Yafang Cheng are members of the editorial board of Atmospheric Chemistry and Physics

**Acknowledgments.**

This work was supported by the National Natural Science Foundation of China (No.42175117 and 495    41877303), Special Fund Project for Science and Technology Innovation Strategy of Guangdong Province (No.2019B121205004), and Guangdong Innovative and Entrepreneurial Research Team Program (No.2016ZT06N263).

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

**Table 1.** Gravimetric densities ($\rho$) and hygroscopicity parameters ($\kappa$) for each component used in the ZSR calculation.

| Compounds | $\rho$ ($kg\ m^{-3}$) | $\kappa$ |
|---|---|---|
| $(NH_4)_2SO_4$ | 1769 | 0.48 |
| $NH_4HSO_4$ | 1780 | 0.56 |
| $NH_4NO_4$ | 1720 | 0.58 |
| $H_2SO_4$ | 1830 | 0.90 |
| SOA | 1400 | -- |
| POA | 1000 | 0 |
| BC | 1600 | 0 |

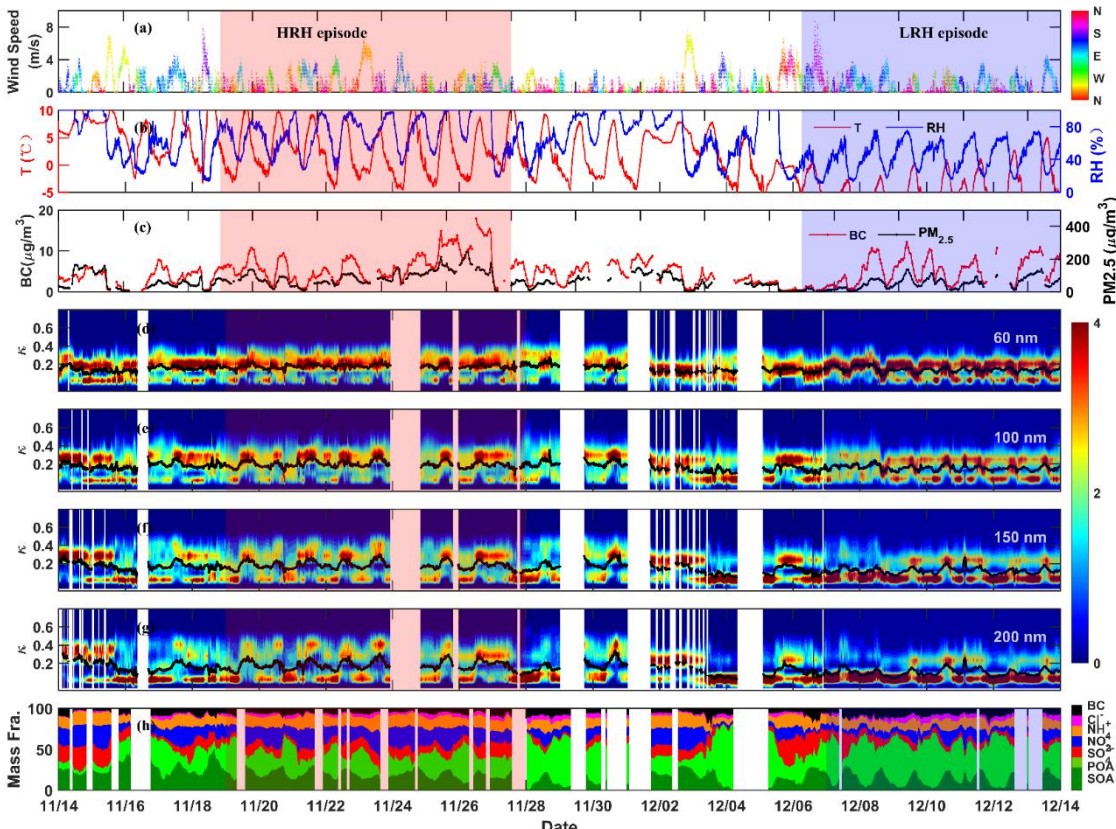

**Figure 1: Time series of (a) wind speed and direction, (b) temperature and relative humidity, (c) PM₂.₅ and BC mass concentrations, (d)-(g) the hygroscopicity parameter (κ) probability density function (κ-PDF) for particles at dry sizes of 60, 100, 150 and 200 nm (the black line is the averaged hygroscopicity parameter κ for particles at each size) and (h) mass fractions of the PM₁ chemical components during this field campaign.**

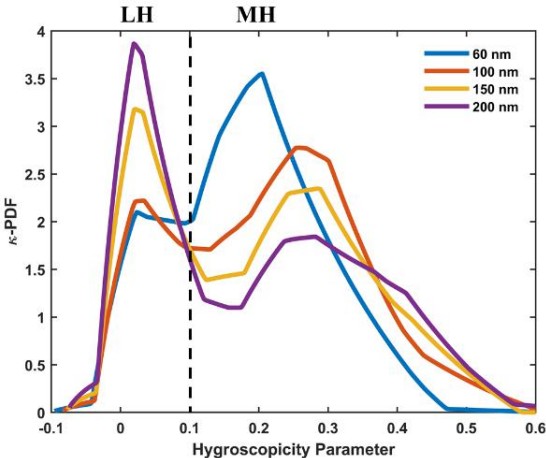

**Figure 2: The average hygroscopicity parameter probability density functions (κ-PDF) for 60 nm, 100 nm, 150 nm and 200 nm sized particles over the entire campaign. Two hygroscopic modes of the particles are defined as: κ < 0.1 as less hygroscopic (LH) mode and κ > 0.1 as more hygroscopic (MH) mode.**

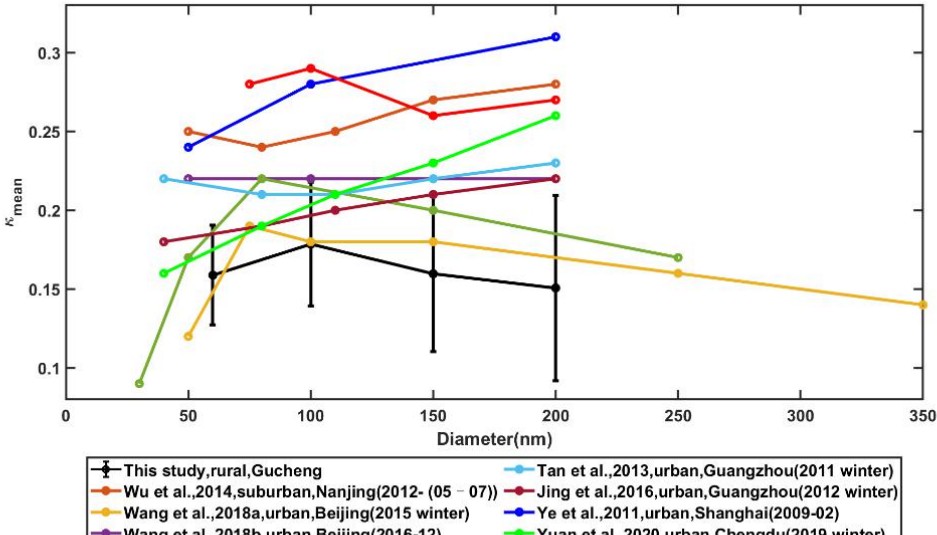

**Figure 3: The ensemble mean hygroscopicity parameter values (κ$_{mean}$) for ambient aerosols of different cities in China.**

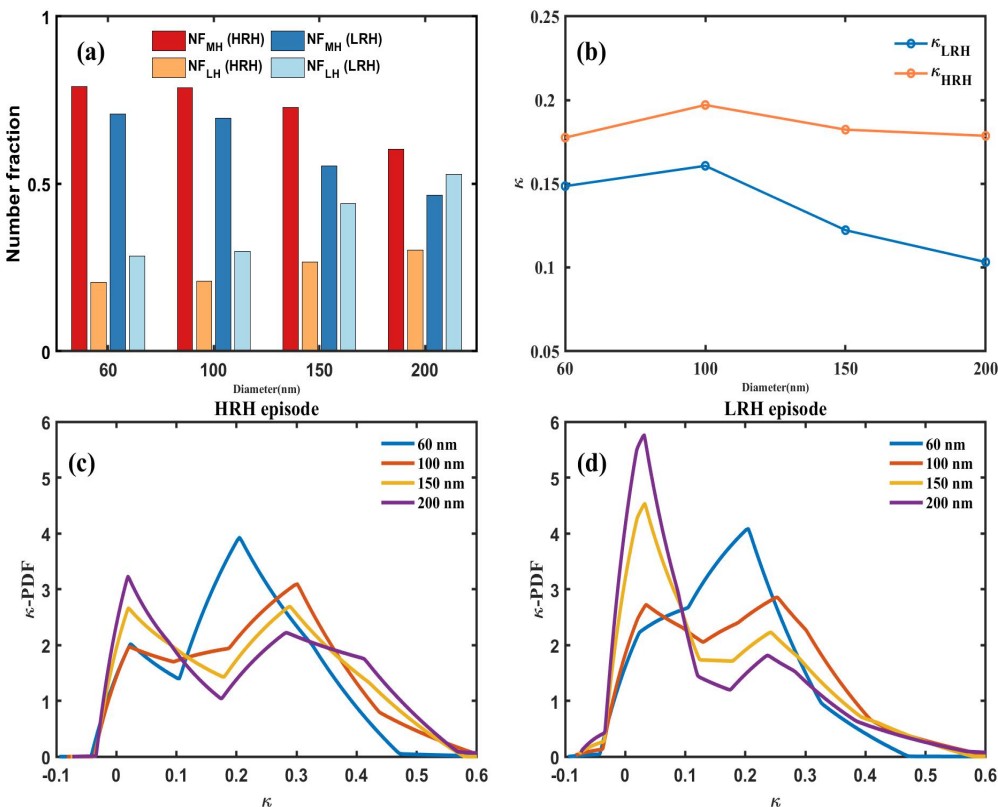

Figure 4: Number fractions of the MH mode and LH mode in particles at 60, 100, 150 and 200 nm, and the average $\kappa$ value of particles at each size under the HRH and LRH episodes.

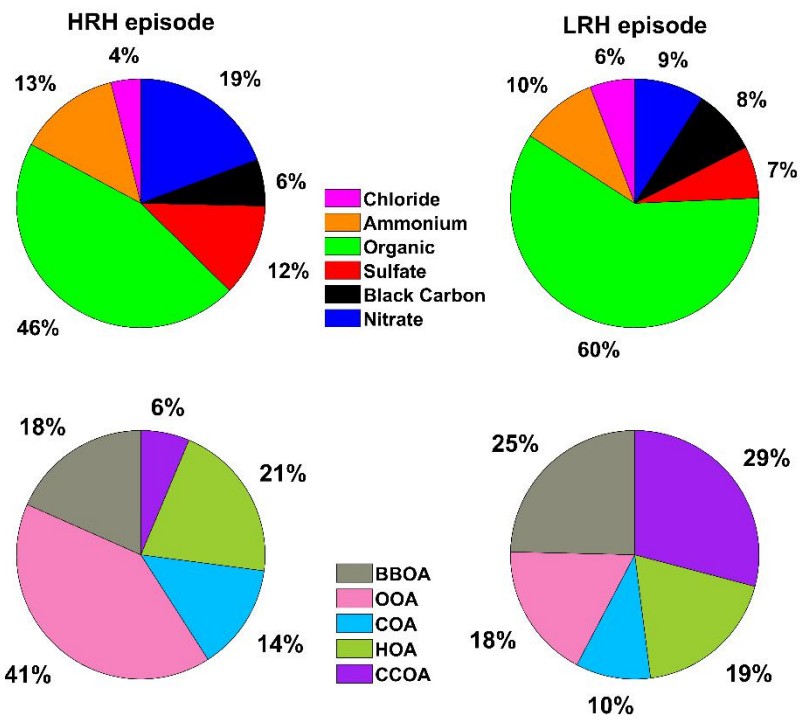

**Figure 5: A comparison of the PM₁ chemical composition during the HRH and LRH episodes. (BBOA: biomass burning organic aerosols, OOA: oxygenated organic aerosols, COA: cooking organic aerosols, HOA: hydrocarbon organic aerosols, CCOA: coal combustion organic aerosols).**

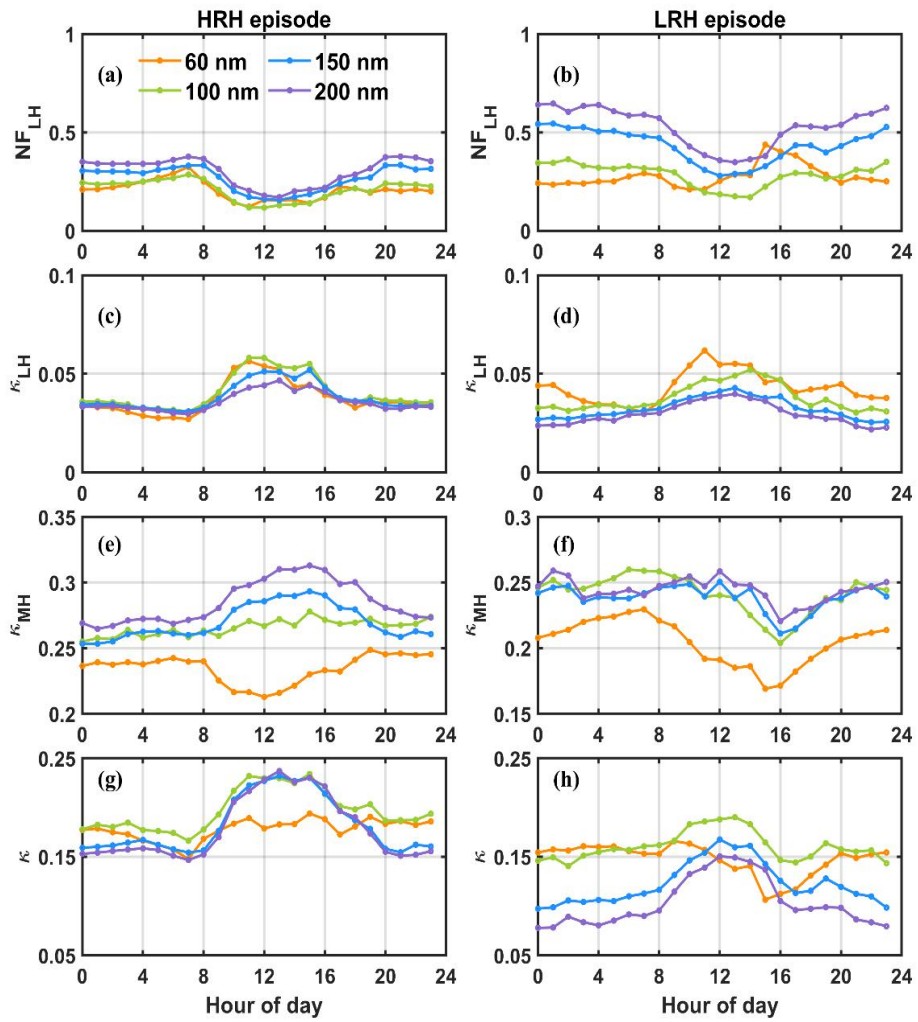

**Figure 6: Diurnal variation of the number fractions and individual *κ* of LH and MH mode particles under these two episodes.**

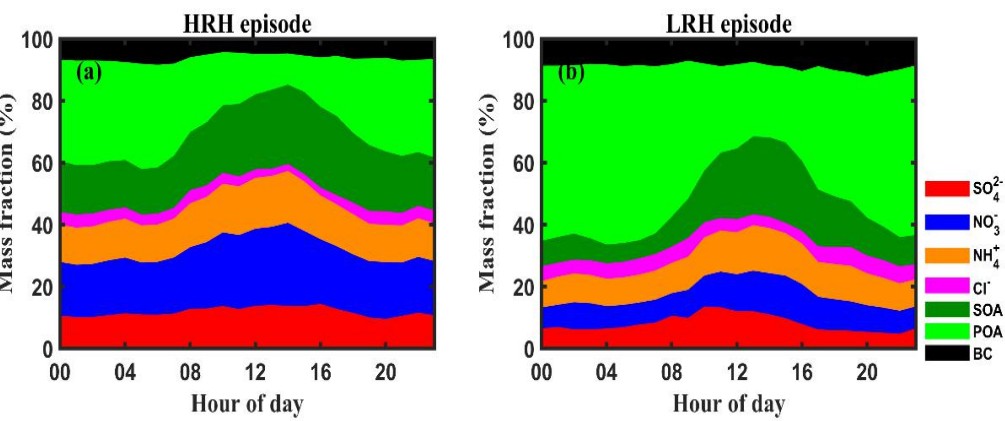

**Figure 7: Diurnal variation of the mass fraction of SO$_4^{2-}$, NH$_4^+$, NO$_3^-$, BC, POA, and SOA in particle phase during the HRH and LRH episodes.**

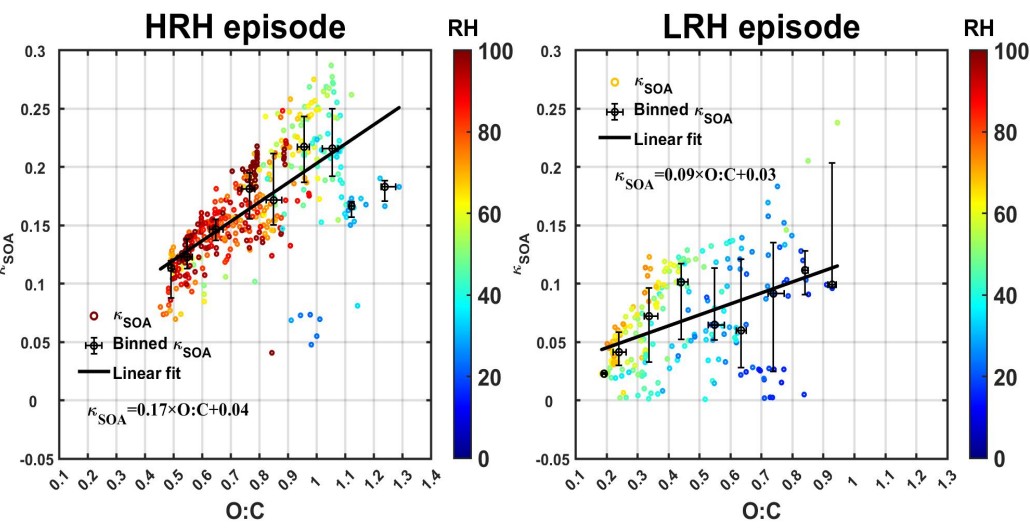

Figure 8: The plots of $\kappa_{SOA}$ vs. O : C ratios during the HRH and LRH episodes. The black line is the fitting to the measured data. The black point is the data that $\kappa_{SOA}$ values were binned by O: C with an increment of 0.1. The color bars indicate RH.

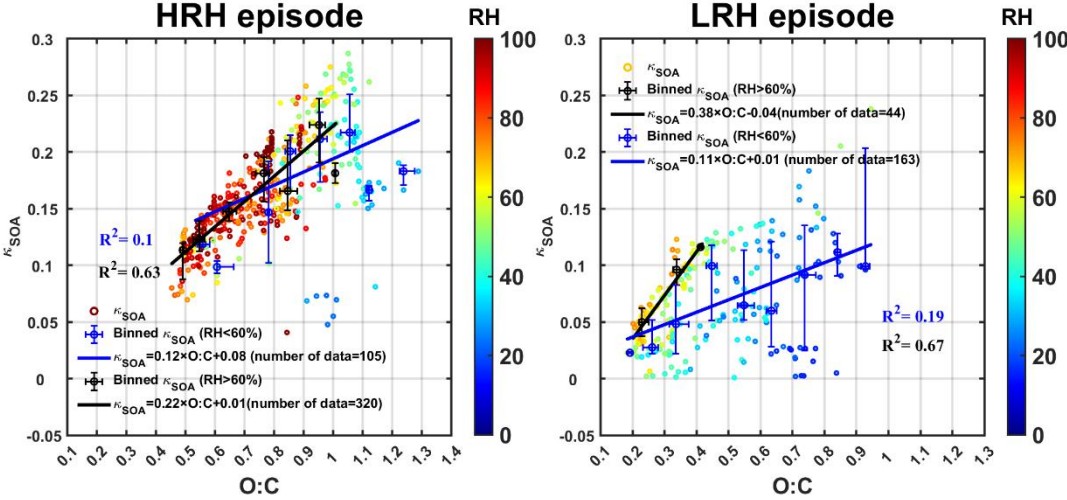

**Figure 9: The plots of $\kappa_{SOA}$ vs. O : C ratios during the HRH and LRH episodes. The black line and blue line are the fitting to the measured data at RH < 60 % and RH > 60 %, respectively. The blue and black point is that $\kappa_{SOA}$ values were usually binned by O: C with an increment of 0.1. The color bars indicate RH.**
