# Peer review of "Measurement report: On the difference of aerosol hygroscopicity between high and low RH conditions in the North China Plain"

_Atmospheric Chemistry and Physics, 2021_

## Author Comment (AC1)

**Response to reviewer #1**

We appreciated referee#1's positive feedback and constructive suggestions which are of great value for improving the quality of our paper. Our point-to-point replies to the referee's comments are listed below.

The manuscript by Shi et al. reports measurements of aerosol hygroscopicity under high and low relative humidity scenarios in the North China Plain. The results show that the kappa values are higher which mean more hygroscopic during high humidity episodes (HRH) than during low humidity episodes (LRH). This distinct difference was attributed to the different chemical composition of the particles under the two scenarios, particularly the O:C ratio, an indicative of the oxidation level. Hygroscopicity is an important aerosol property and hence understanding the influencing factors and the controlling processes is essential to mechanistic understanding the aerosol formation and the climate effects. The paper is well-prepared and can be publishable after the following issues are fully resolved.

1. In section 4.2, the comparison of CCOA and OOA between the HRH and the LRH is qualitatively fine; however, it might be more accurate to consider the significant difference of the organic fraction between the two scenarios, for example, for CCOA, they become 59%*29% (0.17) vs 46%*6% (0.028) and for OOA, the values become closer, i.e., 59%*18% (0.11) vs 46%*41% (0.19).

Response:

Thanks for the specific comment. We added discussions about the difference in organic fractions between these two scenarios in Sect. 4.2: "We can see that the mass fraction of POA in the total organics during the HRH episode was 23 % lower than that during the LRH episode. In particular, CCOA, which is the main component of POA, accounted only for 6 % of the total organics under the HRH episode, but as much as 29 % under the LRH. Considering the distinct content of total organics between the two episodes, the difference in the mass fraction of CCOA in $PM_1$ would be even more significant, being 2.8 % (46 %×6 %) for the HRH and 17% (60 %×29 %) for the LRH. The high level of CCOA in aerosols could be explained that during the LRH episode, the wintertime residential heating was initiated, during which a significant amount of its pollutants, for instance, CCOA, was emitted to the atmosphere (Hua et al., 2018). Moreover, we observed that the mass fraction of OOA in $PM_1$ during the HRH episode was 19% (46 %×41 %), being different from that (11%, e.g., 60 %×18 %) during the LRH, though only slightly higher. Thereby, the varied level of different organics in $PM_1$ could be one of the plausible reasons for the different $\kappa$ in 200 nm between these two episodes."

2. The description (lines 290-300 on p.11) of reaction mechanism for the two periods might need to be noted since there are no more evidences to dig out the formation mechanisms during the two periods. It seems they are just speculated in this case.

Also, the average RH values for both periods should be given and compared to Sun et al. and Yu et al.'s studies. Similar notices should be given when describing Fig. 6e for the mechanisms (lines 327-329) in section 4.3.

Response:

Following the reviewer suggestion, we revised the discussion part of this paragraph. On the other hand, Kuang et al. (2020) only compared the oxidation level of OOA formed through gas-phase or aqueous-phase reaction with the ones generated in laboratory measurements by Sun et al. (2010) and

Ye et al. (2014). However, no comparison in ambient or experimental RH was specified in their works. Hence, the average RH values cannot be compared to the studies by Sun et al. (2010) and Yu et al. (2014), but instead with Kuang et al. (2020). The revised discussion was listed below:

"Kuang et al. (2020) specifically studied the formation mechanism of secondary organic aerosols for current campaign. They found that the daytime OOA formation rates correlated quite well with nitrate formation rates and thus suggested that they probably shared similar formation pathways. They further investigated the formation pathways of nitrate and concluded that during the low RH conditions, nitrate was mainly formed by the gas-phase oxidation of $NO_2$, while at the high RH conditions, both the aqueous-phase processes and gas-phase oxidation dominated its formation. Furthermore, they considered that gas-phase formation of OOA would mainly add mass to the condensation mode of aerosol size distribution, while aqueous-phase formation of OOA may elevate the mass in the droplet mode. After studying the diurnal evolution of aerosol mass distribution for high RH and low RH conditions, together with the indirect evidence from nitrate formation, they concluded that gas-phase formation contributed dominantly to the OOA under the low RH conditions, while at high RH conditions, when aerosol water content was high, aqueous-phase photo-oxidation were mainly responsible for the rapid OOA formation, beside the gas-phase formation. According to their further comparison with laboratory experiments (Sun et al., 2010; Yu et al., 2014), they suggested that the OOA formed through aqueous-phase reactions were normally more-oxidized compared to that formed by gaseous processes. It needs to specify that the average RH values for the HRH and LRH episode during our campaign were 71 % and 43 %, respectively, being quite close to the ones (53 % & 38 %) for the two defined RH conditions in Kuang et al. (2020). Taking all together, this could reasonably to presume that the observed lower $\kappa$ for 200 nm particles under the LHR episode, not only due to the elevated POA fraction in aerosols, but also a different hygroscopic nature of OOA, compared to that under HRH."

For the description of Fig.6e, we revised it as: "During daytime when solar radiation is stronger, higher concentration of atmospheric oxidant was typically observed (Hong et al., 2015). Thus, the associated photochemistry and other aging processes would be more favorable, leading to a more oxidized material condensing or partition into the existing aerosols and thus an elevated aerosol hygroscopicity being expected."

3. The title of section 4.4 needs to be modified to reflect its content. This section describes the correlation between the O/C ratio and the kappa value under the two scenarios. The current title is similar to that of section 4.2. In addition, is there a better way to describe the oxidation state (level) except for the O/C ratio?

Response:

Thanks for this very constructive suggestion. According to the reviewer's suggestion, we used "Relating the SOA hygroscopicity to its oxidation level for the two episodes".

Besides the O/C ratio, the oxidation state (OSc) is a more robust way to describe the oxidation level of organic fractions as it is not influenced by the hydration and dehydration in the atmosphere. Therefore, we calculated the OSc of organics according to Canagaratna et al. (2015) and replotted Figure 8 as shown below. Similar relationship between the SOA hygroscopicity and its oxidation state was

obtained compared to that using the O/C ratio, though the slope of the fitting line is two times larger, which is actually due to OSc = 2×O: C-H: C. As the main conclusion still holds and the O:C ratio was widely adopted in many previous studies, we kept the original figure using the O:C ratio in order to facilitate any possible future comparison.

[Figure]

Figure 1: The plots of $\kappa_{SOA}$ vs. OSc during the HRH and LRH episodes. The black line is the fitting to the measured data. The red point is that $\kappa_{SOA}$ values were binned by OSc with an increment of 0.1. The color bars indicate the measured RH.

**Technical Corrections**

4. Line 35 on p.2, "particles" should be deleted;
Response: We deleted "particles" in the sentence in line 35.

5. Line 68, "which" here is referred to chemical composition or ambient aerosols, it seems ambiguous;
Response:
As suggested by the reviewer, we modified the sentence into: "==Thus, ambient aerosols owing to their different sources and atmospheric processes, may vary greatly in their chemical compositions and thus show significant difference in their hygroscopicity.=="

6. Lines 78-79, "There also remain … more hygroscopic", this is an ill sentence;
Response:
Thanks for the comment. We modified the sentence into: "==There are also some other studies, which focused on the secondary formed or aged aerosols, being typically characterized as more hygroscopic.=="

7. Lines 89, "since decades"? You probably cannot say "since tens of years", right?
Response:
Yes, we used "over the past decade" instead of "since decades".

8. Line 90, "leaded to a fast…"?
Response:
Thank you for your comment, we changed "leaded to a fast…" into: "led to a fast…".

9. Line 94, it is "showed" not "shown";

Response:

Thank you for your comment. We revised "shown" to "showed".

10. Line 97, "extensive efforts to investigated the…"?

Response:

Thanks for the comment. We changed "investigated" into: "investigate".

11. Line 100, I think it is better not to use article before "different sources" here;

Response:

Thanks for the comment. We modified the sentence into: "Thus, the representative feature that how different sources, including both the primary emissions and secondary processes, impact the hygroscopicity of NCP aerosols still remains unclear."

12. Line 224, don't need "the experimental";

Response:

We deleted "the experimental" in the sentence in line 245.

13. Line 237, "this phenomenon" not "this phenomena";

Response:

Thank you for your comment, we changed "this phenomena" into: "this phenomenon".

14. Line 269, "usually considered as mainly from…", as here is a pronoun so it needs an object;

Response:

Thanks for the comment. We modified the sentence into: "usually considered to be mainly from …".

15. Line 288, "by" should be deleted;

Response:

Thanks for the comment, we deleted "by" in the sentence in line 331.

16. Line 343, "contribute" needs a "to" here;

Response:

We added "to" after "contribute" in the sentence in line 470.

17. Line 379, it is better to change "has to be" to "was" here;

Response:

We changed "has to be" to "was" in line 533.

18. Line 392, "marginally increase" should be "marginal increase" as increase is a noun here.

Response:

Thanks for the comment. We modified "marginally increase" to "marginal increase" in line 547.

**References**

Canagaratna, M.R., Jimenez, J.L., Kroll, J.H., Chen, Q., Kessler, S.H., Massoli, P., Hildebrandt Ruiz, L., Fortner, E., Williams, L.R., Wilson, K.R., Surratt, J.D., Donahue, N.M., Jayne, J.T., Worsnop, D.R., 2015. Elemental ratio measurements of organic compounds using aerosol mass spectrometry: Characterization, improved calibration, and implications. Atmos. Chem. Phys. 15, 253–272. https://doi.org/10.5194/acp-15-253-2015

Hua, Y., Wang, S., Jiang, J., Zhou, W., Xu, Q., Li, X., Liu, B., Zhang, D., Zheng, M., 2018. Characteristics and sources of aerosol pollution at a polluted rural site southwest in Beijing, China. Sci. Total Environ. 626, 519–527. https://doi.org/10.1016/j.scitotenv.2018.01.047

Hong, J., Kim, J., Nieminen, T., Duplissy, J., Ehn, M., Äijälä, M., Hao, L.Q., Nie, W., Sarnela, N., Prisle, N.L., 2015. Relating the hygroscopic properties of submicron aerosol to both gas- and particle-phase chemical composition in a boreal forest 11999–12009. https://doi.org/10.5194/acp-15-11999-2015

Kuang, Y., He, Y., Xu, W., Yuan, B., Zhang, G., Ma, Z., Wu, C., Wang, C., Wang, S., Zhang, S., Tao, J., Ma, N., Su, H., Cheng, Y., Shao, M., Sun, Y., 2020. Photochemical Aqueous-Phase Reactions Induce Rapid Daytime Formation of Oxygenated Organic Aerosol on the North China Plain. Environ. Sci. Technol. 54, 3849–3860. https://doi.org/10.1021/acs.est.9b06836

Sun, Y. L., Zhang, Q., Anastasio, C. and Sun, J.: Insights into secondary organic aerosol formed via aqueous-phase reactions of phenolic compounds based on high resolution mass spectrometry, Atmos. Chem. Phys., 10(10), 4809–4822, doi:10.5194/acp-10-4809-2010, 2010.

Yu, L., Smith, J., Laskin, A., Anastasio, C., Laskin, J. and Zhang, Q.: Chemical characterization of SOA formed from aqueous-phase reactions of phenols with the triplet excited state of carbonyl and hydroxyl radical, Atmos. Chem. Phys., 14(24), 13801–13816, doi:10.5194/acp-14-13801-2014, 2014.

---

## Author Comment (AC2)

**Response to reviewer #2**

Thank you for the positive feedback and helpful suggestions. We have addressed the comments and implemented all suggestions in the revised manuscript as detailed below.

The manuscript gives a report of aerosol hygroscopicity (HT-DMA) and composition (ACSM) measurements in the North China Plain. The authors examine data from two periods they identified based on ambient RH. They conclude that the observed difference in aerosol hygroscopicity between these two episodes was due to different chemical composition (specifically O:C ratio) of ambient aerosol particles.

The manuscript is generally written well and is fit for publication after the following issues are addressed.

1. HRH and LRH episodes and Figure 1. While the figure does give a (small and blurry) overview of the conditions during the measurement period, the actual numbers (and statistics) of the RH/T values for these periods should also be presented. Especially if these numbers are used to classify the measurement period into distinct episodes.

Response:

Thanks for this very constructive suggestion. We included more information for the two defined episodes (including both relative humidity and temperature). The following discussion was also added into the line 233 in section 4.1: "To be specific, the average RH during the HRH episode was 71% ± 22%, with an average temperature of 3 °C, while during the LRH episode the average RH was 43% ± 17%, with an average temperature of -6 °C."

2. Figure 1.
- Please make the plot larger and use higher resolution. If this is a limitation of the preprint stage of publication, then that's understandable, but for the final publication it should be more readable.

- Black line in $\kappa$ plots – it's mentioned in the text what it is, but please add a label/description also to the figure itself or the caption.

Response:

Thank you for your comments. We modified Figure 1 with a better resolution, as shown below. Moreover, a description of the black line in Figure 1 of the revised manuscript was also added in its caption.

[Figure]

**Figure 1: Time series of (a) wind speed and direction, (b) temperature and relative humidity, (c) PM₂.₅ and BC mass concentrations, (d)-(g) the hygroscopicity parameter (κ) probability density function (κ-PDF) for particles at dry sizes of 60, 100, 150 and 200 nm (the black line is the averaged hygroscopicity parameter κ for particles at each size) and (h) mass fractions of the PM₁ chemical components during this field campaign.**

3. Page 8, line 220: "*... mode were almost always observed for all sized particles ...*". Please quantify "almost always". Also, Fig. 1 seems to argue against that statement as the LH and MH modes are not continuous in time and have frequent gaps. As an example, LH mode for 100 nm particles during the HRH episode seems to be present (judging by the small plot) about 60% of the time.

Response:

We agree with the reviewer that current statement may not be proper and should be quantified. Therefore, we carefully analyzed the extent of external mixing during the whole campaign. By doing this, we assume that the cases that number fraction of less hygroscopic mode or more hygroscopic mode is less than 0.1 can be considered as internal mixing, where the other cases are external mixing. Based on this assumption, we found that less than 8 % of the time during the whole campaign could be

considered as internal mixing for all four sized particles. Thereby, we revised the statement in line 239 as: "Two distinct modes with $\kappa < 0.1$ as less hygroscopic (LH) mode and $\kappa > 0.1$ as more hygroscopic (MH) mode were mostly observed for all sized particles from the $\kappa$-PDF, indicating that the particles were mainly externally mixed during our measurements."

4. Figure 6: why was NFMH omitted from the plots?
Response:
Thank you for your comments. In our study, we considered our aerosols only consisting two modes: MH mode and LH mode. Therefore, NF of MH was actually 1- NF of LH mode, which is also the reason we omitted it from the plots.

5. Please include a description of and results from particle size distribution measurements mentioned in the manuscript (p. 13, after Eq. 5) and used to justify the use of 200 nm HT-DMA data.
Response:
Thank you for your comment. We plotted the averaged particle mass size distribution for the whole campaign, see the figure below, which we also added into the revised manuscript in the supplement. The discussion in line 492 at p. 14 was also modified accordingly.

"As the bulk chemical composition measured by ACSM may deviate significantly from that of size-resolved ones, we plotted the particle mass distribution of aerosols averaged over the entire campaign, see Fig. S1 in the supplement. From Fig. S1, we found that the mode size of the mass size distribution of aerosols during our experimental campaign was around 390 nm. Thereby, we considered that the bulk chemical composition measured by our ACSM could nearly reflect or at least be close to that of 200 nm particles."

[Figure]

Figure 2: Particle mass size distributions.

6. Figure 8:
- What are the red points with error bars – averages over some range? Please describe.
- The figure caption says the red line is a fit to data. Should it be "black line" instead?
- How much did the ambient RH vary between individual data points on each plot? Looking at Figure 1, the RH had a fairly large diurnal variation. Also, one could almost group the individual

data points and see several trends. If the data points were, for example, colored by the ambient RH, would distinct groups emerge?

Response:

Thanks again for your detailed comments.

1) The red points (black in the revised one) in Fig.8 of the manuscript demonstrate the average $\kappa_{SOA}$ data within a binned O: C with an increment of 0.1.

2) Yes, you are right. It should be the black line and we revised it accordingly.

3) We seriously considered the comment suggested by the reviewer and carefully examined how the ambient RH varied between individual data points. First, we found that during the nighttime of 6th Dec for LRH episode, ambient RH reached to as high as 100 %. These data points with such high RH level should not be considered as LRH condition based on its definition, thus we removed those data from the analysis for LRH episode. Therefore, Fig. 4-8 in the revised manuscript should be and was modified accordingly, which fortunately did not alter any conclusions that were previously obtained. Second, we replotted Fig. 8 of the manuscript by coloring the data points with ambient RH values, as shown in Fig. 3 in current file below. We speculated that the data points with similar RH ranges during both episodes might be emerged into distinct groups as suggested by the reviewer. In order to further confirm this, we divided the data points into two groups: RH < 60 % and RH > 60 % and separately analyzed the relationship between $\kappa_{SOA}$ and the O:C for these two groups, as shown in Fig. 4 below. It has to be noted that the chosen of the threshold at RH of 60 % was based on the intensity of the concentrated colors, which might be arbitrary but still reasonable. We found that the results are interesting and still consistent with our aforementioned conclusion, which we added into the revised manuscript and extended the discussion. The discussion was listed below:

[Figure]

**Figure 3: The plots of $\kappa_{SOA}$ vs. O : C ratios during the HRH and LRH episodes. The black line is the fitting to the measured data. The black point is the data that $\kappa_{SOA}$ values were binned by O: C with an increment of 0.1. The color bars indicate RH.**

==highlight==
"As the ambient RH had a large diurnal variation during our campaign, which implies that low/high RH conditions may also be occur during any individual day of the HRH/LRH episode, we further grouped the data points of Fig. 8 into two categories according to their absolute RH values for these two episodes, as shown in Fig. 9. The threshold at RH of 60 % was set for these two categories due to the RH intensity spread in Fig. 8. At conditions of RH larger than 60 %, the hygroscopicity of SOA
==highlight==

under the HRH and LRH episodes both show a strong O:C-dependency, with the fitting under the LRH being more skewed. At RHs lower than 60 %, the relationship between $\kappa_{SOA}$ and the O:C for these two episodes became even more closer. However, we observed that the absolute value of $\kappa_{SOA}$ still varied between these two episodes, even at similar RH ranges, though their individual behavior towards the variation of O:C was similar. This implies that there still remain differences in these SOAs at different episodes but similar RH conditions, for instance, their chemical composition, indicating that the formation pathways of these SOA or the relevant reaction precursors might still be different under these two episodes. Thus, separation of these two episodes as previously defined was kept as the main conclusion was not altered and distinct groups with respect to their RH will not be merged further."

[Figure]

**Figure 4: The plots of $\kappa_{SOA}$ vs. O : C ratios during the HRH and LRH episodes. The black line and blue line are the fitting to the measured data at RH < 60 % and RH > 60 %, respectively. The blue and black point is that $\kappa_{SOA}$ values were usually binned by O : C with an increment of 0.1. The color bars indicate RH.**

7. Page 14, line 385: "*... RH ... was also quite high ...*". Please quantify "quite high".
Same comment for "*... low RH conditions ...*" on line 394 on the same page.
Response:
Thanks for the comment. We revised the sentences into: "The average RH during their study in Guangzhou was 72 %, being also quite high and thus suggesting potentially similar formation mechanism for secondary organic aerosols, e.g., aqueous-phase photochemistry as speculated previously." and "We found that during Wu et al. (2016)' study, ambient RH was 40 % on average. Thereby, we speculate that the similar characteristic in hygroscopicity of secondary organic aerosols in the NCP of both studies was likely due to the comparable low RH conditions, indicative of similar formation pathways."

**Minor comments**

8. Please check the use of underscore vs. dash throughout the manuscript. Examples include "*HT-DMA_measured*" and "*ACSM_derived*" on page 2.

Response:

Thank you for your comments. We modified the use of underscores and dashes, such as lines 44, 45, 296, 300, 484, 487, respectively.

9. Page 3, line 68: what varies – aerosol or composition?

Response:

To avoid misleading, we revised the corresponding discussion into: *"Thus, ambient aerosols owing to their different sources and atmospheric processes, may vary greatly in their chemical compositions and thus show significant difference in their hygroscopicity."*

10. Page 3, line 83: "*... quite hygroscopicity ...*". Please review language.

Response:

Thanks for the comment. we revised the phrase"... quite hygroscopicity ..." to "... quite hygroscopic ...".

11. Page 6, line 162: missing "r" in "*analyzer*".

Response:

We changed "analyze" to "analyzer".

12. Page 9, line 229: sentence starting with "*Compared with ...*". Please review language and grammar. Currently it reads as if aerosols were somehow obtained and stored.

Response:

Thanks for the comment. To avoid confusion, we revised the corresponding description into: "The results in our study show that the aerosols at current station have the lowest hygroscopicity compared with aerosols in other cities or regions in China. This is more likely due to the largest contribution of organics relative to inorganic species in $PM_1$ at our observational site.".

13. Page 10, line 282: first sentence – please review language and grammar.

Response:

We revised the corresponding description into: "Organic fraction, which is another major component in aerosols, also varies significantly in hygroscopicity due to the exist of numerous and highly diverse organic compounds.".

14. Figure 5: suggest adding the meanings of the acronyms to the figure caption for easier reference.

Response:

Thanks for the comment. We revised the figure caption in Fig. 5 as: "A comparison of the $PM_1$ chemical composition during the HRH and LRH episodes. (BBOA: biomass burning organic aerosols, OOA: oxygenated organic aerosols, COA: cooking organic aerosols, HOA: hydrocarbon organic aerosols, CCOA: coal combustion organic aerosols)."

[Figure]

Figure 5: A comparison of the PM$_1$ chemical composition during the HRH and LRH episodes. (BBOA: biomass burning organic aerosols, OOA: oxygenated organic aerosols, COA: cooking organic aerosols, HOA: hydrocarbon organic aerosols, CCOA: coal combustion organic aerosols).

15. Page 11, line 291: Kuang et al. (2020) isn't on the references list.

Response:

Yes, we added it into the reference list.

16. Beginning of section 4.3: Figure 6 is discussed, but not referenced.

Response:

Thanks for the comment. We modified the sentence into: "To better understand the influence of human activities and secondary formation on the aerosol hygroscopicity of current study on a daily scale, we compared the diurnal variation of the number fractions and individual κ of LH and MH mode particles under these two episodes, as shown in Fig. 6.".

17. Eq. (5) on page 13: some of the text seems very small – were nested subscripts used by accident (from εBC onward)?

Response:

Thank you for your comments. We modified Eq. (5) on page 14 as suggested.

18. Page 14, line 383: "... *moderately hygroscopic*". Please fix.

Response:

Thanks for the comment. We modified "moderate hygroscopic" to "moderately hygroscopic" in the revised manuscript.

19. Page 14, line 392: please review language and grammar.

Response:

Thanks for the reviewer's specific comments. We rephrased the sentence as: "Similarly, Wu et al. (2016), performed in the urban area of the NCP, showed marginal increase in hygroscopicity with the O : C ratio, being approximate to the ones for our LRH aerosols.".

**References**

Kuang, Y., He, Y., Xu, W., Yuan, B., Zhang, G., Ma, Z., Wu, C., Wang, C., Wang, S., Zhang, S., Tao, J., Ma, N., Su, H., Cheng, Y., Shao, M., Sun, Y., 2020. Photochemical Aqueous-Phase Reactions Induce Rapid Daytime Formation of Oxygenated Organic Aerosol on the North China Plain. Environ. Sci. Technol. 54, 3849–3860. https://doi.org/10.1021/acs.est.9b06836

Wu, Z.J., Zheng, J., Shang, D.J., Du, Z.F., Wu, Y.S., Zeng, L.M., Wiedensohler, A., Hu, M., 2016. Particle hygroscopicity and its link to chemical composition in the urban atmosphere of Beijing, China, during summertime. Atmos. Chem. Phys. 16, 1123–1138. https://doi.org/10.5194/acp-16-1123-2016